# Can LLMs Use Relational Transformer Embeddings?

**Francisco Galuppo Azevedo** [1 2]  **Clarissa Lima Loures** [1 2]

## Abstract

Injecting frozen relational-encoder embeddings as soft tokens into a large language model (LLM) is a conceptually appealing fusion strategy: the encoder handles multi-table structure, the LLM handles language and reasoning, and no lossy text serialization is required. We test this hypothesis concretely by injecting embeddings from a frozen Relational Transformer (RT) into Qwen3.5-4B via a learned MLP projection and LoRA adaptation, trained first with supervised fine-tuning (SFT) on chain-of-thought reasoning traces and then with group-based reinforcement learning (GSPO). We evaluate across 10 binary classification tasks on 6 relational databases from RelBench, under four supervision regimes: single-task (ST), within-dataset (WD), cross-dataset (CD), and all-task (ALL). The hybrid model does not consistently outperform standalone RT: it is frequently below random, highly sensitive to serialization format and relational-token budget, and unstable under RL training. We report these negative results and analyze the failure modes, arguing that soft-token fusion requires stronger alignment objectives and schema-aware design before it can serve as a reliable route to relational prediction.

## 1. Introduction

Relational databases encode entity attributes, foreign-key relationships, and temporal histories across interconnected tables (Fey et al., 2023). Predicting entity-level outcomes from this structure is a core industrial ML task (Robinson et al., 2024). Two paradigms have emerged: specialized relational encoders such as the Relational Transformer (Ranjan et al., 2026), which operate directly over the relational graph; and LLMs prompted with serialized views of the data (Wydmuch et al., 2024).

Each paradigm has complementary limitations. Relational encoders produce rich, structure-aware embeddings but cannot leverage pretrained language reasoning. LLMs reason over text but struggle with multi-table structure: serializing a two-hop neighborhood is lossy, exhausts context windows, and discards the typed semantics of foreign-key graphs.

This complementarity motivates a natural fusion hypothesis: inject frozen RT embeddings into an LLM as soft tokens, bypassing serialization entirely. The RT compresses a relational ego-graph into dense embeddings; the LLM contributes semantic reasoning and cross-schema generalization. But whether this actually works is an open empirical question; the closest prior work (Wu et al., 2025) keeps the LLM frozen and uses a schema-specific GNN requiring per-database pretraining, leaving cross-schema transfer untested.

We test this hypothesis by injecting frozen RT embeddings into Qwen3.5-4B via a learned MLP projection and LoRA adaptation (Hu et al., 2022), trained with SFT on chain-of-thought reasoning traces followed by group-based reinforcement learning (GSPO). We evaluate across 10 binary classification tasks on 6 RelBench databases under four supervision regimes: single-task (ST), within-dataset (WD), cross-dataset (CD), and all-task (ALL).

The answer is largely negative. The hybrid does not consistently outperform standalone RT in any regime, is frequently below random, and is highly sensitive to serialization format, relational-token budget, and training initialization. Soft-token fusion is not a plug-and-play route to better relational prediction.

Our contributions are as follows:

1. A concrete RT-to-LLM soft-token fusion architecture for RelBench binary prediction.
2. Evaluation across four supervision regimes, from single-task to fully cross-dataset transfer.
3. Evidence that the hybrid does not consistently outperform standalone RT and is often unstable.
4. A failure analysis across serialization, relational-token budget, attention masking, and training initialization.

[1]Kunumi Institute, Brazil [2]Universidade Federal de Minas Gerais, Belo Horizonte, Minas Gerais, Brazil. Correspondence to: Francisco Galuppo Azevedo <franciscogaluppo@dcc.ufmg.br>.

*2nd ICML Workshop on Foundation Models for Structured Data (FMSD @ ICML 2026)*, Seoul, South Korea, 2026. Copyright 2026 by the author(s).

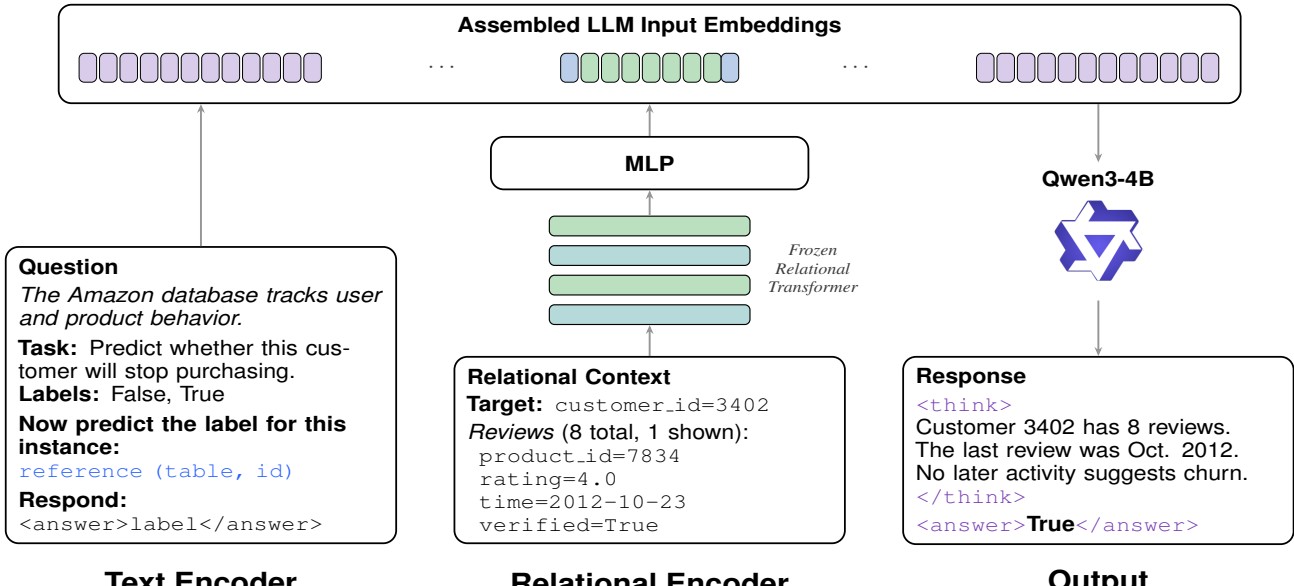

*Figure 1.* Overview of the proposed architecture. A natural-language task prompt, while the entity's relational ego-graph is encoded by a frozen Relational Transformer. The resulting relational embeddings are projected by an MLP into the Qwen3.5-4B embedding space and inserted as soft tokens to the text sequence. The LLM then generates a reasoning trace and emits the binary prediction in answer format.

## 2. Background

### 2.1. Relational Transformer

The Relational Transformer (RT) (Ranjan et al., 2026) is a cross-schema foundation model architecture for relational data. For each target entity, RT constructs a context window via bounded-width BFS over the primary–foreign key graph, enforcing temporal constraints to prevent leakage. The resulting cell tokens are encoded by a transformer with four structured attention masks: column, feature, neighbor, and full attention. RT is pretrained leave-one-database-out on RelBench and achieves strong zero-shot transfer to unseen schemas and tasks. It is our primary baseline and the source of the frozen embeddings we inject into the LLM.

### 2.2. LLMs over Relational Data

Two lines of work apply LLMs to RelBench prediction. Wydmuch et al. (2024) serialize each entity's two-hop neighborhood as nested JSON and query a frozen LLM with in-context examples; performance is competitive with relational deep learning but sensitive to prompt design and context length. Wu et al. (2025) propose Rel-LLM, which injects GNN embeddings as soft tokens into a frozen LLM via a learned projection. The LLM is never trained to use the injected tokens and the GNN requires per-database pre-training, leaving cross-schema transfer untested. Our work differs on both axes: the encoder is a cross-schema foundation model and the LLM is trained via LoRA to interpret the injected relational tokens.

### 2.3. Multimodal Fusion of Foundation Models

Our two-stage training recipe follows BioReason (Fallahpour et al., 2025) and BioReason Pro (Fallahpour et al., 2026): SFT on chain-of-thought reasoning traces to warm up the LLM's use of the injected modality, followed by group-based reinforcement learning (GSPO) (Zheng et al., 2025) with a binary correctness reward. GSPO estimates advantages from completions sampled per prompt, using mean-only normalization (Liu et al., 2025) and asymmetric clipping bounds from DAPO (Yu et al., 2025), aligning the importance-correction unit with the reward at the sequence level. We apply this recipe to relational tabular data, with RT embeddings as the non-language modality.

## 3. Method

### 3.1. Problem Setup

We study binary node classification over relational databases. Each task is defined by a target entity type drawn from a RelBench database (Robinson et al., 2024), a natural-language task description $d$, and a binary label $y \in \{0, 1\}$. The model receives the entity $e$ alongside $d$; the relational context of $e$ is encoded from its ego-graph, a bounded-width subgraph obtained by BFS up to $k$ hops over the primary–foreign key graph, with temporal constraints to prevent leakage. Performance is measured by AUROC on held-out test instances from RelBench's temporal splits. We evaluate on 10 binary classification tasks across 6 databases.

**Flat Tokens**

projected rel. embeddings

text · *continuous vectors* · text

**Hierarchical Tokens (`json`)**

interleaved text tokens & embeddings

text · `{` `field1` `:` `[` `]` `,` `field2` `:` `[` `]` `,` `linked_ents` `:` `[` `{` `field3` `:` `[` `]` `}` `]` `}` · text

*structured by field and entity*

*Figure 2.* Relational-token serialization strategies. In the Flat Tokens format, projected RT embeddings are inserted as a continuous soft-token prefix between delimiter tokens, with no explicit textual structure. In the hierarchical `json` format, relational embeddings are interleaved with field, entity, and punctuation tokens.

### 3.2. Multimodal Architecture

Our model fuses a frozen RT with a parameter-efficiently adapted LLM by treating RT embeddings as continuous soft tokens (Figure 1). For each target entity, the RT encodes its ego-graph into $L = 256$ relational embeddings of dimension $d_{\mathrm{rt}} = 256$. A two-layer MLP with Tanh activation projects each embedding into the Qwen3.5-4B hidden dimension of 2560. The projected embeddings are inserted into a reserved span in the prompt using a `json` serialization format, interleaving soft tokens with textual field and punctuation tokens that expose the schema hierarchy (Figure 2). The LLM attends over the full mixed sequence with a standard causal mask (Figure 3). The RT encoder is frozen throughout; the projection layer and LoRA adapters (Hu et al., 2022) ($r = 16, \alpha = 32$) are the only trained parameters. Serialization format, context size, and attention mask are ablated in Section 4.2.

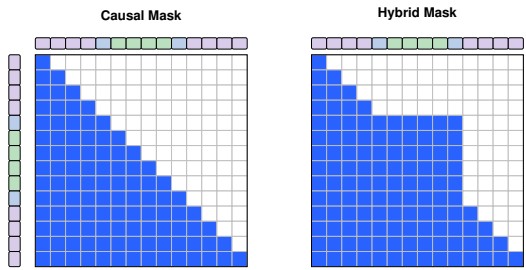

**Causal Mask**

**Hybrid Mask**

*Figure 3.* Attention masks for mixed text–relational input sequences. The causal mask applies standard left-to-right attention over both text tokens and projected relational tokens. The hybrid mask preserves causal attention outside the relational span but allows bidirectional attention within the relational-token block.

### 3.3. Training Pipeline

**SFT warm-up.** We fine-tune on 7,869 training examples derived from 300 base samples per task across all 10 tasks, each augmented into up to three style variations (narrative, analytical, concise) by a teacher LLM (Claude Sonnet 4.6) and filtered for noise (Table 1, Appendix B). Each example trains the model to generate a natural-language description of an entity's relational neighborhood conditioned on its injected RT embeddings, teaching the LLM to interpret relational tokens before any task-specific signal is introduced, analogously to BioReason Pro (Fallahpour et al., 2026).

*Table 1.* SFT with Variations Dataset Statistics

| Task | Inst. | Tokens | Tok/Inst |
|---|---|---|---|
| amazon/item-churn | 883 | 901K | 1020.5 |
| amazon/user-churn | 867 | 869K | 1002.7 |
| avito/user-clicks | 782 | 792K | 1013.3 |
| avito/user-visits | 769 | 772K | 1004.6 |
| f1/driver-dnf | 606 | 794K | 1310.1 |
| f1/driver-top3 | 571 | 917K | 1605.7 |
| hm/user-churn | 890 | 963K | 1081.9 |
| stack/user-badge | 737 | 623K | 845.8 |
| stack/user-engage | 869 | 923K | 1062.6 |
| trial/study-outcome | 895 | 908K | 1014.9 |
| **Total** | **7,869** | **8,464K** | **1075.6** |

Training is performed jointly across all tasks; the resulting checkpoint initializes all subsequent GSPO runs.

**GSPO.** we run GSPO independently per task–regime combination. The reward is 1 if the predicted label matches the ground truth, 0 otherwise; a format reward of 1 is added if the output contains valid `<think>` and `<answer>` blocks. We sample $G=4$ completions per prompt and train for 2 epochs over 100 instances with learning rate $10^{-6}$. Full hyperparameters are in Appendix A.

## 4. Experiments

### 4.1. Setup

We evaluate three hybrid variants, **SFT** (SFT only), **GSPO** (no SFT warm-up), and **SFT+GSPO** (full two-stage pipeline), against standalone **RT** as the primary baseline. All models are evaluated by AUROC on RelBench temporal test splits across four regimes: ST (same task at train and test, upper bound), WD (different task, same schema), CD (different schema entirely), and ALL (all tasks jointly).

### 4.2. Ablation Study

We ablate three design axes on two tasks (driver-dnf and study-outcome) using binary training reward as the metric. Results are in Table 2. Rather than identifying a winning configuration, the ablation reveals that performance is highly

sensitive to design choices that should be irrelevant if the LLM were robustly using the injected relational tokens.

| Axis | Config. | Data | Task | Format | Time |
|------|---------|------|------|--------|------|
| rel-attn mask | bidir⋆ | f1 | 0.073 | 0.279 | 114 |
| | | trial | 0.223 | 0.707 | 115 |
| | **causal** | f1 | +0.047 | +0.482 | 106 |
| | | trial | −0.028 | −0.104 | 104 |
| serialization | flat⋆ | f1 | 0.073 | 0.279 | 114 |
| | | trial | 0.223 | 0.707 | 115 |
| | **json** | f1 | +0.632 | +0.718 | 42 |
| | | trial | +0.359 | +0.263 | 53 |
| context size | 64⋆ | f1 | 0.073 | 0.279 | 114 |
| | | trial | 0.223 | 0.707 | 115 |
| | **256** | f1 | +0.070 | +0.479 | 110 |
| | | trial | +0.144 | +0.084 | 106 |
| | 1024 | f1 | +0.213 | +0.446 | 139 |
| | | trial | +0.114 | +0.080 | 132 |

*Table 2.* Ablation results (1 seed). Non-default rows show $\Delta$ over the default. The ⋆ marker denotes the default value for each axis; colored entries indicate the selected configuration.

**Attention mask.** Causal attention improves task reward on rel-f1 (+0.047) but slightly hurts rel-trial (−0.028). The inconsistency across two tasks already suggests the model is not learning a stable strategy for attending to relational tokens, the right mask depends on the task rather than being universally better.

**Serialization.** Switching from flat tokens to `json` yields large gains on both tasks (+0.632 on rel-f1, +0.359 on rel-trial). This is the largest effect in the ablation, and it is concerning: if the LLM were using the continuous relational embeddings effectively, the textual scaffolding around them should be a minor factor. The strong sensitivity to serialization suggests the LLM is relying heavily on the text tokens rather than the soft tokens themselves.

**Context size.** Increasing $L$ from 64 to 256 improves task reward modestly (+0.070 on rel-f1, +0.144 on rel-trial); increasing further to $L=1024$ adds little while increasing memory and training time. We adopt $L=256$ as a practical compromise, but the diminishing returns above 64 suggest the LLM is not extracting proportionally more information from additional relational tokens.

Together, these results paint a consistent picture: the hybrid model is fragile along axes that a well-functioning fusion architecture should be robust to. The LLM appears to lean on textual structure rather than learning to decode the injected relational representations.

### 4.3. Main Results

Table 3 reports mean AUROC by regime; full per-task results are in Appendix Table 6. Standalone RT is the strongest model in every regime and no hybrid configuration closes the gap on average.

| Model | ST | WD | CD | ALL |
|-------|------|------|------|------|
| RT | **71.9** | — | **69.7** | — |
| SFT | | 48.3 | | |
| GSPO | 51.1 | 53.5 | 54.9 | 61.4 |
| SFT+GSPO | 50.7 | 48.2 | 52.2 | 54.4 |

*Table 3.* Mean AUROC by supervision regime. RT has no WD or ALL results by design. SFT is regime-independent (one checkpoint per task). GSPO ALL mean is inflated by two outlier tasks; see Appendix Table 6 for per-task detail.

In the ST regime, SFT collapses to near-random (48.3), and both GSPO (51.1) and SFT+GSPO (50.7) are nearly identical: SFT warm-up provides no benefit. In WD, SFT+GSPO (48.2) is weaker than GSPO-only (53.5), the opposite of the expected ordering. In CD, RT retains 69.7, only marginally below ST, while the hybrid models fall well short. In ALL, GSPO reaches 61.4 but this is driven by two outlier tasks (user-clicks=81.5, user-visits=68.9); the remaining eight tasks average 57.9.

The hybrid beats RT only on its two weakest tasks: user-churn on rel-hm ST (GSPO 64.8 vs RT 63.3) and study-outcome ST (GSPO 55.9 vs RT 54.6). On RT's strong tasks these margins do not appear.

Training curves for rollout length, task reward, and format reward are reported in Appendix Figure 4; format reward saturates early across all configurations, confirming that task-reward differences reflect prediction quality rather than format-following failures.

## 5. Conclusion

We tested whether frozen RT embeddings injected as soft tokens into an LLM improve relational prediction. Across 10 tasks, 6 databases, and four supervision regimes, the answer is largely no. The hybrid does not consistently outperform RT, is frequently below random, and is highly sensitive to serialization format. Soft-token fusion is not a plug-and-play route to better relational prediction.

Our evaluation has limitations: binary classification only, a single 4B-parameter LLM, 100 GSPO training samples per task–regime combination, and single-seed results. Future work should investigate stronger alignment objectives, projection layers conditioned on schema metadata rather than a schema-agnostic MLP, and whether larger training budgets or larger LLMs change the picture.

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

# A. Hyperparameters

| Hyperparameter | Value |
|---|---|
| Base model | Qwen3.5-4B |
| LoRA rank ($r$) | 16 |
| LoRA alpha ($\alpha$) | 32 |
| LoRA dropout | 0.05 |
| LoRA target modules | q_proj, k_proj, v_proj, o_proj |
| Optimizer | AdamW ($\beta_1{=}0.9$, $\beta_2{=}0.999$, $\varepsilon{=}10^{-8}$) |
| Learning rate | $2 \times 10^{-5}$ |
| LR schedule | linear |
| Per-device batch size | 2 |
| Epochs | 2 |
| Precision | bf16 |
| Seed | 42 |
| Relational serialization | json |
| Causal rel-attn | true |
| $L$ (seq_len) | 256 |

*Table 4.* SFT stage hyperparameters.

| Hyperparameter | Value |
|---|---|
| Base model | Qwen3.5-4B (SFT init for SFT+GSPO; base for GSPO-only) |
| LoRA rank ($r$) | 16 |
| LoRA alpha ($\alpha$) | 32 |
| LoRA dropout | 0.05 |
| LoRA target modules | q_proj, k_proj, v_proj, o_proj, embed_tokens |
| Optimizer | AdamW ($\beta_1{=}0.9$, $\beta_2{=}0.999$, $\varepsilon{=}10^{-8}$) |
| Learning rate | $10^{-6}$ |
| LR schedule | linear |
| Weight decay | 0 |
| Gradient clipping | 1.0 |
| Per-device batch size | 2 |
| Gradient accumulation steps | 4 |
| Effective batch size ($B$) | 8 |
| Group size ($G$) | 4 |
| Unique samples per step ($B/G$) | 1 |
| KL penalty ($\beta$) | 0.04 |
| $\varepsilon_{\text{low}}$ | 0.2 |
| $\varepsilon_{\text{high}}$ (DAPO) | 0.28 |
| Max completion tokens | 1024 |
| Max model length (vLLM) | 12,288 |
| vLLM GPU memory fraction | 0.2 |
| Epochs | 2 |
| Training samples per task | 100 |
| Precision | bf16 |
| Seed | 42 |
| Relational serialization | json |
| Causal rel-attn | true |
| $L$ (seq_len) | 256 |
| Hardware | $1\times$ H200 |

*Table 5.* GSPO stage hyperparameters.

# B. SFT Data Generation

**Ego-graph extraction.** For each entity in the training split of every task, we extract a structured ego-graph from the relational database: the center entity's attributes plus all directly reachable rows from related tables, up to a configurable hop depth and row limit. The result is serialized as a YAML-structured dump that preserves table names, column names, and type values, serving as the structured input to the teacher model.

**Text generation with three style variations.** A teacher model (Claude Sonnet 4.6 via the Anthropic API) is prompted with the ego-graph and a style instruction appended to the base ego-graph prompt (Appendix C). Three variation styles are defined in `template/variations.yaml`: *narrative* (temperature 0.3, a flowing description emphasizing the entity's trajectory and context); *analytical* (temperature 0.0, leading with statistics and precise numbers); and *concise* (temperature 0.0, 2–3 sentences capturing the single most salient fact). Each base sample thus yields up to three candidate training examples.

**Noise filtering.** Four noise types are detected and removed: *yaml_echo* (the model echoed the structured YAML input instead of generating natural prose); *refusal* (the model refused or claimed insufficient data); *meta_comment* (the model described the task or the data format instead of the entity); and *too_short* (fewer than 20 words). YAML echoes are first auto-stripped by removing lines that match the input structure; if the remaining text is still noisy, the variation is dropped entirely. After filtering, each sample's surviving variations are flattened into independent training examples: a sample with three clean variations contributes three separate entries to the final JSONL.

**Output format.** Each training example is stored as a three-turn conversation in standard chat format: a *system* message defining the assistant's role, a *user* message containing the task prompt, and an *assistant* message containing the generated description. This format is consumed directly by the SFT trainer, which computes cross-entropy loss only over the assistant turn.

## C. Prompts

**System message** You analyze database entities and predict labels based on their relational context.

---

**Task Prompt**

{database_description}

You are given structured data about instances from this database. Some instances have known labels, and you must predict the label for the final instance.

Task: {task_description}
Possible labels: {label_options}

{labeled_instances}

Now predict the label for this instance:

{unlabeled_instance}

Respond with your answer in this exact format: ¡answer¿label¡/answer¿

---

**Ego-graph JSON structure**

```json
{
  "attributes": {  },
  "related_tables": {
    "<table_name>": {
      "n_total": <total edges>,
      "rows": [ <up to max_path_shown rows> ]
    }
  }
}
```

**SFT Reasoning Trace Prompt**

Structure your response exactly as follows:
<think>
A fluid, concise narrative showing the step-by-step deduction a domain expert would follow when analyzing this data instance — examining each piece of evidence, identifying patterns, and building toward the final description.
</think>
Your final description here.

Do not add any text before the opening <think> tag.

{database_description}

You will receive structured data about a specific instance from this database.
Your task is to write a natural, informative description based on this data.

Guidelines:
- Write in clear prose without bullet points or lists
- State facts naturally as if explaining to someone unfamiliar with the database
- Do not mention that you're reading from structured data, a YAML file, or an ego graph
- Do not use phrases like "based on the data" or "according to the information provided"
- Only include information that is explicitly present or can be directly inferred from the data
- Use specific numbers and details when available
- Focus on the most salient and interesting patterns in the data

<documentation>
{ego_graph_documentation}
</documentation>

<data>
{ego_graph_dump}
</data>

## D. Full Results

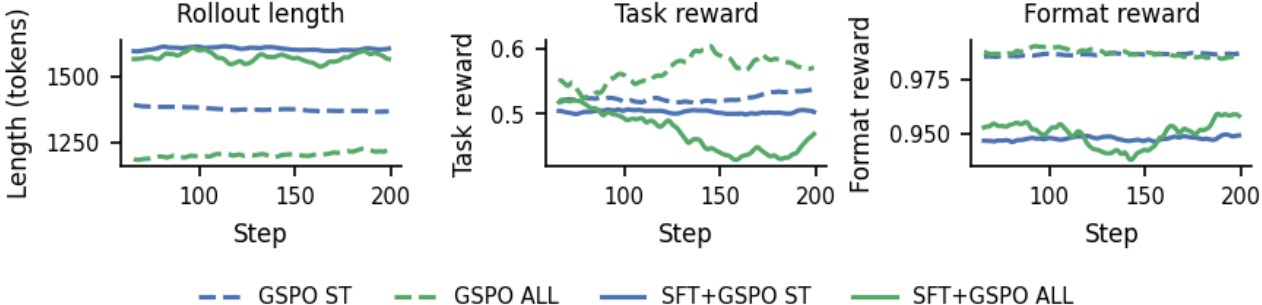

*Figure 4.* Averaged training curves for rollout length, task reward and format reward.

| Task | Regime | RT | SFT | GSPO | SFT+GSPO |
|------|--------|-----|------|------|----------|
| driver-dnf | ST | **81.2** | | 51.5 | 43.6 |
| | WD | — | 44.0 | 61.1 | 44.1 |
| | CD | **81.2** | | 61.0 | 41.4 |
| | ALL | — | | 56.8 | 43.6 |
| driver-top3 | ST | **89.3** | | 46.3 | 54.9 |
| | WD | — | 59.1 | 58.7 | 56.9 |
| | CD | **89.3** | | 71.0 | 58.3 |
| | ALL | — | | 52.9 | 50.6 |
| user-churn[†] | ST | **66.1** | | 48.3 | 58.2 |
| | WD | — | 54.5 | 57.8 | 54.9 |
| | CD | 64.0 | | 54.0 | 57.7 |
| | ALL | — | | 65.7 | 65.1 |
| item-churn | ST | **73.3** | | 52.1 | 45.5 |
| | WD | — | 45.3 | 50.7 | 45.5 |
| | CD | 70.9 | | 48.2 | 50.8 |
| | ALL | — | | 49.0 | 51.8 |
| user-visits | ST | 62.6 | | 52.8 | 46.8 |
| | WD | — | 40.8 | 42.2 | 41.4 |
| | CD | 61.8 | | 52.0 | 49.2 |
| | ALL | — | | **68.9** | 56.4 |
| user-clicks | ST | 60.9 | | 44.4 | 41.8 |
| | WD | — | 37.4 | 38.7 | 40.1 |
| | CD | 59.5 | | 48.2 | 41.6 |
| | ALL | — | | **81.5** | 43.0 |
| user-engagement | ST | **86.9** | | 58.2 | 46.2 |
| | WD | — | 44.3 | 49.8 | 40.2 |
| | CD | 75.7 | | 43.0 | 43.0 |
| | ALL | — | | 55.6 | 57.9 |
| user-badge | ST | **81.1** | | 36.3 | 67.8 |
| | WD | — | 62.1 | 68.8 | 62.5 |
| | CD | 80.1 | | 61.5 | 73.8 |
| | ALL | — | | 63.3 | 70.4 |
| user-churn[‡] | ST | 63.3 | | **64.8** | 47.7 |
| | CD | 62.8 | 45.9 | 58.7 | 52.2 |
| | ALL | — | | 60.7 | 50.4 |
| study-outcome | ST | 54.6 | | 55.9 | 54.2 |
| | CD | 51.8 | 49.1 | 51.3 | 54.1 |
| | ALL | — | | **59.7** | 54.8 |

*Table 6.* Main results in AUROC. Color indicates performance: red below-random performance, gray near-random performance, yellow moderate performance, green strong performance. Bold marks the highest value for each task. [†]rel-amazon; [‡]rel-hm.

