# OpenReview forum: "Can LLMs Use Relational Transformer Embeddings?"
_ICML.cc/2026/Workshop/FMSD — FMSD @ ICML 2026 Poster_

### Official Review · Reviewer_Ebcc · 2026-05-13
**Interesting Negative Result, but Failure Analysis Is Not Yet Convincing**

**Rating:** 4
**Confidence:** 4

**Review:**

## Summary

This paper tests whether frozen Relational Transformer embeddings can improve LLM-based relational prediction when injected as soft tokens. The model projects RT embeddings into the Qwen embedding space and trains the projector plus LoRA adapters using SFT followed by GSPO. The main result is negative: the hybrid model usually underperforms standalone RT and appears fragile across design choices.


## Strengths

**The work is highly relevant to the workshop.**
The submission directly addresses foundation models for structured and relational data. It studies relational foundation-model embeddings, LLM fusion, cross-task prediction, and RelBench-style evaluation.

**The paper asks an important and timely question.**
Combining a relational encoder with an LLM is a natural idea for foundation models on structured data. The paper directly tests whether RT embeddings can be injected into an LLM as soft tokens for relational prediction.

**The negative result is valuable.**
The paper is useful because it does not simply assume that adding an LLM improves performance. It shows that this appealing fusion strategy performs poorly compared with standalone RT, which is an important cautionary result for the community.


## Areas for Improvement

**The paper shows poor performance, but not a convincing cause of failure.**
The authors argue that the LLM is not effectively using the injected RT embeddings, but the evidence is mostly indirect. The serialization ablation is suggestive, but it does not convincingly show that the soft tokens are ignored.

**The paper needs stronger diagnostic baselines.**
The paper should include controls such as random RT embeddings, zeroed embeddings, shuffled embeddings across examples, JSON/text scaffolding without RT embeddings, text-only LLM prompting, and a direct classifier on frozen RT embeddings. These would clarify whether the model is using the relational embeddings or instead relying on textual scaffolding, label priors, task-specific artifacts, the projection layer, or the training objective.

**The SFT objective is weakly aligned with the final task.**
SFT trains the model to generate natural-language descriptions of relational contexts, while the final task is binary classification. This may teach fluent relational summaries rather than decision-relevant use of embeddings. The authors should ablate the SFT phase more carefully, including comparisons to label-supervised SFT.

**The choice of GSPO is under-motivated.**
The paper borrows a recent group-based RL method, but does not justify why GSPO is needed for this setting. Other RL methods may perform better and should be compared or discussed.

## Detailed Comments
The main additions I would like to see are diagnostic controls, SFT ablations, and RL ablations to better understand why the setup does not work well. I would also encourage the authors to narrow the framing: the current experiments show that this specific pipeline underperforms RT, but they do not establish a broad conclusion about whether LLMs can use relational transformer embeddings injected as soft tokens.

## Justification of Score
The question is timely and highly relevant to the workshop, and I appreciate the attempt to report a negative result. However, the negative result is not yet convincing enough as a general finding: the experiments show that this particular implementation underperforms RT, but they do not rule out other fusion designs, alignment objectives, or training setups for using relational transformer embeddings as soft tokens.

---

### Official Review · Reviewer_KAoC · 2026-05-21

**Rating:** 6
**Confidence:** 3

**Review:**

### Summary
This paper investigates whether the relational transformer (RT) embeddings can be injected as continuous soft tokens into an LLM to improve prediction for relational databases.  The paper uses a frozen RT encoder, with an MLP projection into qwen3.5-4b's embedding space and the LoRA adaptation and GSPO-style reinforcement learning. Their evaluation considers binary classification tasks over 6 databases, and consider single task, withing dataset, cross dataset and all-task regimes. The finding of the paper is negative. Their RT-to-LLM model does not consistently improve over the standard RT baseline, and often is close to near random prediction and also quite sensitive to formats for serialization, and various other training details.

### Strengths
1. The paper investigates whether a relational foundation-model embeddings can be used as a modality for LLM-based reasoning over structured databases. In my opinion, despite the negative results, this is an important direction to investigate.
2. The empirical scope is sufficiently broad.

### Areas for Improvement
1. Although I really liked the reporting of this negative result, a number of important questions came to my mind. It seems that the experiments use one seed and only 100 GSPO training samples per task-regime, which might be quite limiting. Also, all the experiments are conducted with one LLM architecture, which might be another limiting factor. My main issue with that is that whether the reported negative results extend to any soft token fusion approach is not addressed. It would be good to have more generality while reporting a negative result.
2. The paper needs controlled experiments to isolate the cause of failure. In particular, the current experiments do not clearly separate failures of the fusion idea from failures of the specific training recipe. What could result in this negative result? Does RT embeddings have no value to add over the LLM representation or a specific training recipe is not suitable for this setting?

### Detailed Comments
1. The paper shows that JSON serialization gives large improvements over flat tokens. Without a text-only JSON baseline, it is hard to know whether the RT embeddings contribute anything beyond.
2. Authors can replace RT embeddings with random vectors, shuffled embeddings, or embeddings from entities with mismatched labels. If performance does not change much, that would strongly show that the LLM is not effectively using the relational embeddings.
3. Since the model is described as unstable, reporting multi seed results is valuable.

### Justification of Score
I believe the overall direction of the paper, and its negative reporting to be valuable. However, I find the analysis behind this negative result to be lacking. I gave a weak accept but I am borderline between weak accept and weak reject. Please do not hesitate to override my opinion.

---

### Official Review · Reviewer_gEUh · 2026-05-21

**Rating:** 5
**Confidence:** 3

**Review:**

Summary
- This paper proposes integrating embeddings from a relational transformer into Qwen3.5-4B through MLP projection and LoRA adaptation, and evaluates the approach under both SFT and GSPO training settings. Experiments across six databases show mixed performance signals.

Strengths
- Comprehensive evaluation across four supervision regimes.
- Useful failure analysis covering serialization methods, token budget trade-offs, and training strategy ablations, which may inform future research directions.

Areas for Improvement
- The study is limited in scope, focusing primarily on binary prediction tasks.
- The negative results, where the hybrid approach underperforms the standalone relational transformer, raise concerns about the effectiveness of the proposed methodology.